# Cell Wall-Mediated Antifungal Activity of the Aqueous Extract of *Hedera helix* L. Leaves Against *Diplodia corticola*

**DOI:** 10.3390/antibiotics13121116

**Published:** 2024-11-22

**Authors:** Christina Crisóstomo, Luara Simões, Lillian Barros, Tiane C. Finimundy, Ana Cunha, Rui Oliveira

**Affiliations:** 1Department of Biology, School of Sciences, University of Minho, Campus de Gualtar, 4710-057 Braga, Portugal; christina98@live.com.pt (C.C.); accunha@bio.uminho.pt (A.C.); 2Centre of Molecular and Environmental Biology (CBMA), Department of Biology, University of Minho, Campus de Gualtar, 4710-057 Braga, Portugal; luaraasimoes@gmail.com; 3Centre for the Research and Technology of Agro-Environmental and Biological Sciences (CITAB), University of Trás-os-Montes e Alto Douro, 5000-801 Vila Real, Portugal; 4Interdisciplinary Centre of Marine and Environmental Research, University of Porto, Terminal de Cruzeiros do Porto de Leixões, Avenida General Norton de Matos, 4450-208 Matosinhos, Portugal; 5CIMO, LA SusTEC, Instituto Politécnico de Bragança, Campus de Santa Apolónia, 5300-253 Bragança, Portugal; lillian@ipb.pt (L.B.); tiane@ipb.pt (T.C.F.)

**Keywords:** *Hedera helix* water extract, antifungal mechanism of action, anti-phytopathogen activity, *Diplodia corticola*, *Saccharomyces cerevisiae* fungal model

## Abstract

**Background/Objectives:** Cork oak forests have been declining due to fungal pathogens such as *Diplodia corticola*. However, the preventive fungicides against this fungus have restricted use due to the deleterious effects on human health and the environment, prompting the need for sustainable alternatives. Here, we describe the antifungal activity of an aqueous extract of *Hedera helix* L. leaves (HAE) against *D. corticola* and the possible mechanism of action. **Results/Methods:** The chemical analysis revealed compounds like the saponin hederacoside C, quinic acid, 5-*O*-caffeoylquinic acid, rutin, and glycoside derivatives of quercetin and kaempferol, all of which have been previously reported to possess antimicrobial activity. Remarkable in vitro antifungal activity was observed, reducing radial mycelial growth by 70% after 3 days of inoculation. *Saccharomyces cerevisiae* mutants, *bck1* and *mkk1*/*mkk2,* affected the cell wall integrity signaling pathway were more resistant to HAE than the wild-type strain, suggesting that the extract targets kinases of the signaling pathway, which triggers toxicity. The viability under osmotic stress with 0.75 M NaCl was lower in the presence of HAE, suggesting the deficiency of osmotic protection by the cell wall. **Conclusions:** These results suggest that ivy extracts can be a source of new natural antifungal agents targeting the cell wall, opening the possibility of preventing fungal infections in cork oaks and improving the cork production sector using safer and more sustainable approaches.

## 1. Introduction

Since the middle of the 20th century, fungal diseases affecting crops have grown in number and severity, and now constitute a significant threat to the safety of the world’s food supply [1]. It is estimated that pathogen infections are responsible for losses in agricultural production of about 20 to 40% [2]. *Diplodia corticola* is an endophytic fungus of the Botryosphaeriaceae family that can affect plants of different ages [3,4]. This fungus is one of the most dangerous pathogens to *Quercus* species [5]. Cork oak (*Quercus suber* L.; Félix et al. [6]) forests have been attacked by *D. corticola* in several Mediterranean countries, including Italy, Morocco, Portugal, Spain, and Tunisia [5]. Although the infection mechanism of this fungus remains unknown, an opportunistic behavior is recognized, like that shown by some other cork oak fungal pathogens [5,7]. Sunken cankers on the collar, trunk, and branches are the most frequent signs of *D. corticola* infection in cork oak trees [5].

Fungicides have been used successfully to prevent fungal diseases and maintain crop quantity and quality [8]. The mode of action varies depending on the fungicide used, with the best-known mechanisms being effects on plasma membrane integrity, the microtubule cytoskeleton, and the inhibition of mitochondrial respiration. Most of the synthetic fungicides target specific enzymes, such as azoles [1], which bind to 14 α-demethylase (CYP51) and inhibit the synthesis of ergosterol, a key component for maintaining membrane fluidity and permeability [9]. Usually, because of their low cost, long-lasting stability, and their broad antifungal activity against several fungal infections, such as mildews and rusts of grains, fruits, vegetables, and ornamentals, powdery mildew in cereals, berry fruits, vines, and tomatoes, azoles are the preferred treatment for fighting pathogenic fungi in agriculture [9,10]. Also, echinocandins, lipopeptide molecules, have been widely used due to their capacity for inhibiting the enzyme 1,3-β-D-glucan synthase, which is present in fungal but not in mammalian cells. This inhibition weakens the cell wall, leading to fungal cell lysis and probably death [11]. With regard to the treatment of cork oak infections by *D. corticola*, there are two fungicides that have proved effective: triophanate-methyl and carbedazim. However, their application is limited due to the possible negative impacts on human health and the environment [12].

Despite fungicides’ advantages in the fight against fungal infections, using them on plants will also harm other fungi, especially aquatic fungal species, and contribute to the disruption of ecosystems [13]. In addition, the excessive use of pesticides has negative effects on humans and contributes to the development of resistant strains [14,15]. The World Health Organization (WHO) estimates that pesticide poisoning causes 300,000 fatalities annually worldwide [16]. The negative effects caused by pesticides have increasingly led to the search for alternatives, namely biopesticides [17,18]. These control agents are natural substances or obtained from living organisms that have a shorter life cycle than synthetic pesticides, which reduces their persistence in the field. They also have a more limited spectrum because of their high specificity, necessitating the employment of many biopesticides to combat various phytopathogenic agents, and a slower killing rate [17,19,20]. Important advantages include the fact that normally, these natural compounds are less toxic to farmers and consumers, and usually only affect the target pest and phylogenetically close organisms [20,21]. Also, although the shorter life cycle may reduce exposure time and the need for more applications, it also avoids accumulation of noxious molecules in the environment. Regarding plant extracts, it is well known that plants produce several secondary metabolites that, despite not being directly connected to basic main processes like growth, development, and reproduction, are crucial for plant interaction with and adapting to the environment and have numerous uses in medicine, cosmetics, and nutrition [22,23]. Some of the secondary metabolites, such as tannins, terpenoids, saponins, alkaloids, and flavonoids, possess several biological activities including antifungal properties [24]. Numerous studies indicate that these substances could be an important source for the formulation of novel biopesticides (review by Marutescu et al. [19]).

*Hedera helix* L., also known as ivy or English ivy, is a climbing plant that belongs to the Araliaceae family [25,26]. This plant species plays an important role in traditional medicine due to the antifungal, anthelmintic, molluscicidal, antileishmanial, and anti-mutagenic activities of the saponins present in the leaves [26,27]. To the best of our knowledge, the antifungal activity of English ivy ethanolic extract was first reported in 2017 against several plant pathogens, such as *Aspergillus niger*, *Botrytis cinerea*, *Botrytis tulipae*, *Penicillium gladioli*, and *Sclerotinia sclerotiorum*, with minimum inhibitory concentrations lower than the synthetic drug fluconazole [28]. The abundance of secondary compounds in *H. helix* as well as the antifungal activity already demonstrated against some phytopathogenic fungi led to the interest in studying extracts of this plant against the fungus *D. corticola*, which affects cork oaks, the source of cork, a high-valued product with a variety of applications.

Because the conventional fungicides for *D. corticola* have restricted use [12], plant extracts such as *H. helix*-derived products could have potential as biodegradable natural antifungals and to replace or reduce the use of fungicides currently used to combat *D. corticola* infections. The objective of this work is to find natural products with an effective antifungal activity to protect *Q. suber* against *D. corticola* in order to improve cork production without using synthetic fungicides. An *H. helix* water extract was investigated for antifungal activity against *D. corticola* and the toxicity mechanism was investigated by chemical characterization and viability assays with mutant strains of the fungal cellular model *Saccharomyces cerevisiae* affected in ergosterol biosynthesis and the cell wall, the targets of synthetic fungicides. Assays of osmotic sensitivity in the presence of the extract were also performed in the fungal model and *D. corticola* to investigate targeting to the cell wall.

## 2. Results

### 2.1. HAE Chemical Analysis

A comprehensive chemical analysis of *H. helix* aqueous extract (HAE) was performed to determine its phenolic composition and other key constituents, providing insights into the main compounds present in the extract. The concentrations of these compounds were quantified as follows: the total phenolic content in the extract was 16.1 ± 0.3 mg/g, with 6.1 ± 0.2 mg/g of total flavonoids, including flavones, flavanones, and flavonols. The extract is also rich in saponins (2.21 ± 0.01 mg/g). Among the compounds identified, quinic acid was found in the highest amount (3.42 ± 0.04 mg/g), followed by 5-*O*-caffeoylquinic acid (1.87 ± 0.01 mg/g; Table 1). Other notable compounds include di-caffeoylshikimic acid (1.74 ± 0.03 mg/g), eriodictyol-*O*-glucuronide (1.87 ± 0.05 mg/g), and quercetin-3-*O*-rutinoside (rutin) at 1.72 ± 0.02 mg/g. Additionally, the extract contains various flavonoid glycosides, such as apigenin-8-*C*-hexoside (0.26 ± 0.01 mg/g) and apigenin-6-*C*-hexoside-8-*C*-rhamnoside (0.31 ± 0.01 mg/g), as well as polyphenols like kaempferol-3-*O*-rutinoside (1.15 ± 0.03 mg/g) and isorhamnetin-*O*-rutinoside (0.82 ± 0.04 mg/g). Notably, the saponin hederacoside C was found among the components of the extract, which is characteristic of *H. helix* leaves.

The results concerning the phenolic composition of HAE, peak characteristics, tentative identification, and quantification of each compound are presented in Table 1. In order to characterize the individual compounds present in the aqueous extract, their tentative identification was performed based on their retention time (Rt), the wavelength of maximum absorption in the UV–Vis spectra (λmax), deprotonated ion ([M-H]^−^), and fragmentation pattern (MS^2^). In total, eleven compounds were identified, including phenolic acids (**peaks 1**, **2**, **4**, and **6**), flavonoids (**peaks 3**, **5**, **7**, **8**, **9**, and **10**), and one saponin (**peak 11**). Compounds **1, 2**, and **8** were identified by comparison with commercial standards.

**Peak 1**, quinic acid, was identified by its precursor ion at *m*/*z* 191 and a fragment at *m*/*z* 161, which is consistent with the literature for this compound. The observed fragmentation represents the loss of a water molecule (18 Da), which is characteristic of quinic acid, and which was also confirmed with a commercial standard. **Peak 2** was identified as 5-*O*-caffeoylquinic acid, with a precursor ion at *m*/*z* 353 and a characteristic fragment at *m*/*z* 191, resulting from the loss of a caffeoyl group (162 Da). This loss is common in chlorogenic acid derivatives and was also confirmed using a commercial standard. **Peak 4** was tentatively identified as yunnaneic acid D, with a molecular ion at *m*/*z* 539 and fragments at *m*/*z* 353 and 191. The fragmentation suggests the loss of a caffeoyl unit (162 Da), followed by the release of various fragments due to successive losses of 44 Da (CO_2_) and 197 Da (dihydroxyphenyl-lactic acid). These characteristics are similar to those described for yunnaneic acid, an identity that was tentatively associated with this compound. Di-caffeoylshikimic acid was identified as **peak 6**, with a molecular ion at *m*/*z* 497 and fragments at *m*/*z* 353, 337, 335, 211, and 179. The loss of 162 Da, resulting in a fragment at *m*/*z* 353, corresponds to the elimination of two caffeoyl units. Regarding the flavonoids, two flavones, one flavanone, one flavone, one flavonol, and one *O*-methylated flavonol were identified. **Peak 3**, tentatively identified as apigenin-8-*C*-hexoside, presented a precursor ion at *m*/*z* 431 and fragments at *m*/*z* 341 and 311 Da, indicative of the loss of a hexosyl unit (90 Da), followed by the subsequent loss of CO (30 Da). This fragmentation sequence confirms the *C*-glycosidic linkage of apigenin (−169 Da) to the sugar. **Peak 5**, apigenin-6-*C*-hexoside-8-*C*-rhamnoside, presented a precursor ion at *m*/*z* 577 and fragments at *m*/*z* 487, 473, 457, and 353. The fragment at *m*/*z* 487 indicates the loss of a rhamnose unit (90 Da), and the fragment at *m*/*z* 473 reflects the loss of a glucose unit (120 Da). The final fragment at *m*/*z* 353 suggests the presence of the apigenin aglycone (−169 Da) after the complete loss of the sugars. **Peak 7** was identified as eriodictyol-*O*-glucuronide, with a molecular ion at *m*/*z* 463 and a fragment at *m*/*z* 287, corresponding to the eriodictyol aglycone after the loss of the glucuronide group (176 Da). This loss is typical of a glucuronidated flavonoid. Quercetin-3-*O*-rutinoside, also known as rutin, was identified as **peak 8**, with a molecular ion at *m*/*z* 609 and a fragment at *m*/*z* 301, characteristic of quercetin after the loss of the disaccharide rutinoside (308 Da); this peak was also identified by taking into account the commercial standard. **Peak 9**, kaempferol-3-*O*-rutinoside, was identified, with a molecular ion at *m*/*z* 593 and a fragment at *m*/*z* 285, corresponding to the kaempferol aglycone after the loss of the rutinoside group (308 Da). **Peak 10** was attributed to isorhamnetin-*O*-rutinoside, with a molecular ion at *m*/*z* 623 and a fragment at *m*/*z* 315, characteristic of isorhamnetin after the loss of the disaccharide rutinoside (308 Da). Finally, **peak 11** was identified as hederacoside C, a saponin with a molecular ion at *m*/*z* 1219 and fragments at *m*/*z* 423 and 367, common in triterpenoid saponins. The fragments indicate the loss of several glycosidic units, reflecting the complexity of the saponin structure.

To contextualize these findings, it is important to compare them with the phenolic profiles of ethanol-based extracts. As described by Rosca-Casian et al. (2017) [28], the aqueous extracts of *H. helix* show differences in the identified phenolic compounds, particularly in terms of concentration and the profile of saponins and flavonoids. The choice of the aqueous extract in this study was based on its greater compatibility with physiological conditions in living organisms, as well as the fact that aqueous methods are often used in traditional practices and herbal therapies. Additionally, the aqueous extract demonstrated promising activity against *D. corticola*, possibly due to the presence of bioactive compounds such as quinic acid, 5-*O*-caffeoylquinic acid, and hederacoside C, which are known for their antimicrobial properties.

### 2.2. Antifungal Activity of HAE Against Diplodia corticola

The antifungal activity of HAE was assessed in vitro against *D. corticola* to pre-evaluate the potential upscale application in farming. Different concentrations of the extract were incorporated in a solid medium, and the antifungal activity was estimated by measuring the mycelium diameter after 3 and 6 days of incubation. Exposure to the lowest concentration tested (50 µg/mL HAE) significantly reduced the mycelium diameter when compared with the negative control after 3 days of incubation (*p* ≤ 0.05), but after 6 days, no significant differences were found (Figure 1A). For all the other concentrations, the mycelium diameter decreased significantly after 3 (*p* < 0.0001) and 6 days (*p* < 0.0001; Figure 1A) of incubation when compared with the negative control. The diameter of mycelia was used to calculate the percentage of growth inhibition (Figure 1B). Remarkably, the highest inhibition was attained with 1000 µg/mL of HAE at the third day (~70%), with 500 and 1500 µg/mL inducing slightly non-significant lower values (65% and 64%, respectively). At 100 µg/mL, the inhibition was still as high as 50%.

Interestingly, inhibitions decreased from 3 to 6 days of incubation, suggesting the adaptation of the fungus and/or degradation of the extract. Nevertheless, after 6 days of incubation, inhibition is still remarkable, as percentages were still considerably high, ranging from 37% with 100 µg/mL HAE to 47% with 1000 µg/mL HAE (Figure 1A,B). To our knowledge, extracts of this plant have not yet been tested against *D. corticola*, so these results seem to be promising for the future application of *H. helix* extracts or its isolated phytochemicals in the control of infections caused by this phytopathogenic fungus.

### 2.3. Mechanism of Action of HAE

In order to investigate the mechanism of action of the antifungal activity of HAE, viability tests were performed with *S. cerevisiae* wild-type strain BY4741 and mutant strains such as *erg2*, *bck1*, and *mkk1*/*mkk2.* Budding yeast was selected as a fungal model organism for the mechanistic investigations because it is a well-known organism in terms of cell biology, biochemistry, and physiology; because the mutant strains available are affected in important cellular processes and structures and are likely to be targets of antifungal agents; and because of the validated laboratory techniques due to the extensive research with this experimental model [29,30]. The *erg2* mutant is devoid of C-8 sterol isomerase, the enzyme that catalyzes the conversion of fecosterol into episterol, which is a crucial step in the biosynthesis of the plasma membrane sterol ergosterol. This mutant is affected in the plasma membrane with higher sensitivity to salt stress and cationic drugs [31]. The *bck1* and *mkk1*/*mkk2* mutants are affected in kinases of the cell wall integrity pathway, a signaling pathway triggered by cell wall stress so that genes involved in cell wall remodeling are activated. These mutants typically show high sensitivity to cell wall stress and to high osmolarity (reviewed by Fuchs and Mylonakis 2009 [32]).

The approach with the mutant strains is based on the comparison of cytotoxicity in strains affected by cellular processes or structures with the wild-type strain, so that potential molecular or pathway targets can be disclosed. Accordingly, if a mutant strain shows no signs of cytotoxicity, this might imply that the pathway or structure affected by the mutation is a potential target of the extract. So, in carrying out this study, it was hypothesized that if a mutant strain shows greater resistance to the extract compared to the wild-type strain, this would indicate that the bioactive compound(s) would probably bind to the protein encoded by the mutated gene or to another interacting protein.

Cells from exponentially growing cultures were incubated with different concentrations of HAE and viability was assessed using CFU with time. As depicted in Figure 2A, with the wild-type strain (BY4741), the control and the lowest concentration of HAE tested (10 µg/mL) showed similar viability, indicating that the extract did not affect fungal cells. However, higher concentrations of HAE dramatically affected viability. After 30 min of incubation, the proportion of viable cells in the presence of 50 µg/mL of HAE decreased to values below 50% (*p* < 0.001; Figure 2A) and kept decreasing until 60 min. In the presence of 75, 100, or 250 µg/mL of HAE, the viability was already around 0% after 30 min of incubation (*p* < 0.0001; Figure 2A).

The biosynthesis of ergosterol, an essential and exclusive component of fungal plasma membranes, involves the participation of 30 enzymes known as Erg proteins [33,34]. To study whether HAE could target ergosterol biosynthesis, we used the mutant strain *erg2*, which lacks the active C-8 sterol isomerase enzyme involved in the conversion of the intermediary fecosterol into episterol. Tested under the same conditions, the viability of this mutant strain (Figure 2B) was similar to the wild-type strain (Figure 2A)**,** suggesting that the extract does not target ergosterol biosynthesis.

As ergosterol or its biosynthesis pathway are unlikely to be targeted by the extract, we evaluated whether the target could be the cell wall using mutant strains affected in *BCK1* or in the redundant pair of genes *MKK1* and *MKK2*. These genes are involved in the same protein kinase signaling pathway, i.e., the cell wall integrity (CWI) signaling pathway. This pathway consists of a three-part MAPK phosphorylation cascade, made up of the MEK kinase Bck1, the redundant MEKs Mkk1 and Mkk2, and the MAPK Slt2, which, through successive phosphorylations, amplifies and relays the signals produced on the cell surface to downstream destinations such as gene-encoding enzymes involved in cell wall repair [35]. Mutants affected in this pathway typically have phenotypes of cell wall deficiency and are more sensitive to cell wall stress [36]. The viability of the *bck1* mutant, as expected, increased with time (Figure 2C) in a similar way to *erg2* (Figure 2B) and the wild-type strain (Figure 2A). This behavior was similar when cells were exposed to 10 µg/mL; however, unlike the wild type and *erg2* strains, when cells were exposed to 50 or 75 µg/mL of HAE, viability did not change throughout the whole experiment. Only with the two higher concentrations tested, a significant decrease to 50% (100 µg/mL; *p* < 0.05; Figure 2C) and 0% (250 µg/mL; *p* < 0.001; Figure 2C), respectively, was observed after 30 min. With respect to the double mutant strain *mkk1*/*mkk2*, the viability of the negative control and in the presence of 10 µg/mL of HAE also increased over time. The same behavior was observed when cells were exposed to 50 µg/mL of HAE; however, unchanged viability was observed when cells were exposed to 75 or 100 µg/mL of HAE (Figure 2D). A clear reduction in viability was observed in the double mutant when compared with the control, but only with 250 µg/mL of HAE, reaching approximately 0% after 30 min (*p* < 0.0001; Figure 2D). The results with mutants affected in CWI are in clear contrast with the results obtained with the wild-type and the *erg2* mutant strain. The remarkably lower sensitivity when *BCK1* or *MKK1* and *MKK2* genes are non-functional suggests that HAE targets the cell wall or the signaling pathway that controls cell wall remodeling.

To investigate the hypothesis of the cell wall being a target of HAE, cells from cultures grown in the presence of the extract were tested for osmotic sensitivity. Cell wall-deficient cells are typically osmotically sensitive; therefore, this assay would assess if HAE, in some way, affects cell wall integrity or its ability to protect cells against osmotic stress. Non-exposed and exposed cells to 10 µg/mL of HAE displayed similar viability, regardless of the presence or the concentration of NaCl (*p* > 0.05; Figure 3). As expected, the activity of 50 µg/mL of HAE is visible in the absence of NaCl, with nearly 50% viability (Figure 2A and Figure 3), but when cells were exposed to increasing salt concentrations, a clear progressive decrease in viability was observed. This effect was more pronounced in 0.75 M of NaCl, in which a significant decrease of viability was observed, suggesting that the toxicity of the extract is aggravated under osmotic stress and might be related with the cell wall. The results were similar for 60 min incubation with HAE.

Based on the results of increased osmotic sensitivity in the presence of HAE, the influence of the extract on osmotic sensitivity was also investigated in *D. corticola*. This approach would allow us to show that HAE might also target the cell wall in this phytopathogen, as was found for the model *S. cerevisiae*. Radial growth was similar when comparing PDA containing 0.4 M NaCl with PDA plates. A higher concentration, such as 0.8 M of NaCl, affected growth so, 0.4 M was the concentration selected for the experiments. As depicted in Figure 4A,B, the presence of 0.4 M of NaCl increased the growth inhibitory effect of 50 µg/mL of HAE (*p* ≤ 0.001) and of 500 and 1500 µg/mL of HAE (*p* ≤ 0.0001). These results are clear indications that, as in *S. cerevisiae*, HAE affects the integrity of the *D. corticola* cell wall.

## 3. Discussion

The chemical analysis of the aqueous extract of *H. helix* revealed the presence of phenolic compounds, namely, flavonoids, phenolic acids, and triterpene saponins. Other studies have also revealed the presence of some of the compounds identified in our work in *H. helix* extracts, such as quinic acid [37], 5-*O*-caffeoylquinic acid (chlorogenic acid) [38], kaempferol-3-*O*-rutinoside [39,40], quercetin-3-*O*-rutinoside (rutin) [41], and hederacoside C [42,43]. These compounds have been described in the literature as having relevant biological activities. Typically, the triterpene saponins, making up 2.5–6% of the leaves, are the primary active compounds. Among these, hederacoside C constitutes the largest portion, ranging from 1.7% to 4.8% [42,43], which is in line with the result found in our work, where the concentration of hederacoside C was of 2.21 ± 0.01 mg/g of leaves. Previous studies have showed the antioxidant, antimicrobial, and anti-inflammatory capacity of this compound [44]. The most abundant compound present in HAE was quinic acid. This cyclohexanecarboxylic acid can be found in many plants and has been shown to have activities such as antioxidant, antidiabetic, anticancer, antimicrobial, antiviral, antiaging, and protective effects on tetrahydropapaveroline (THP)-induced cell damage in rat C6 glioma cells, and anti-nociceptive and analgesic effects [45]. The phenolic compound 5-*O*-caffeoylquinic acid has some demonstrated biological properties, namely antioxidant, chelating, antimicrobial, antiviral, and anti-carcinogenic properties, and is used as an ultraviolet filter. Regarding antifungal activity, this has already been demonstrated against *Cryptococcus* and *Candida* species [46]; therefore, it is tempting to suggest that these compounds may also be related to the antifungal activity of *H. helix* against *D. corticola*.

Polyphenols like rutin, present in HAE (Table 1), have been indicated as contributors for plant extract antifungal activities, such as *H. helix* [28] and *Moringa oleifera* extracts [47], further reinforcing their antifungal activity in HAE. In addition, chlorogenic acids were also detected in the chemical analysis. Quinic acid (tetrahydroxy-cyclohexane carboxylic acid) and caffeic acid (3,4-dihydroxy cinnamic acid) are conjugates that make up their parent structure. A broad family of chlorogenic acids arises as a result of isomers and epimers in the cyclohexane portion and replacements at the aromatic ring [48]. The antifungal potential of lipophilic chain chlorogenic acid derivatives is already recognized. Some of these compounds, initially, were designed to mimic the pharmacophores of echinocadin, an already mentioned well-known inhibitor of fungal 1,3-β-*D*-glucan synthase that is essential for the synthesis of the cell wall component of fungi [49]. The most common of the chlorogenic acids is the 5-*O*-caffeoyl quinic acid (5-CQA).

To the best of our knowledge, the antifungal activity of *H. helix* aqueous extracts against *D. corticola* has never been studied; so, this work showed very promising results for combating *D. corticola*, a serious cork oak infection agent. The in vitro results demonstrated the strong activity of ivy extract against the *D. corticola*, with a remarkable 70% and 45% inhibition after 3 and 6 days of incubation, respectively (Figure 1B). The antifungal activity of *H. helix* has been shown before, with a 50% ethanol extract against phytopathogenic fungi such as *A. niger*, *B. cinerea*, *B. tulipae*, *F. oxysporum*, *P. gladioli* and *S. sclerotiorum* [28]. So, the results of this study are in line with previous reports on the *H. helix* content of antifungal metabolites that have activity against fungi affecting important crops.

The plasma membrane and the cell wall were investigated as potential targets of compounds of HAE using *S. cerevisiae* as the model organism. This choice was made on the basis that these cell structures have unique features common to all fungi and are targeted by common synthetic fungicides, namely, the inhibition of ergosterol synthesis for the plasma membrane by azoles and inhibition of synthesis of the cell wall polymer component beta-(1,3)-*D*-glucan by echinocandins. While the *erg2* mutant strain displayed similar sensitivity as the wild-type strain, indicating that HAE does not target ergosterol biosynthesis (Figure 2B), *bck1* and *mkk1*/*mkk2* mutants were more resistant to the extract compared to the wild-type strain (Figure 2C,D). This suggests that at least one of the targets of the aqueous extract of *H. helix* might be the cell wall. In yeast, there are several factors that can lead to the activation of the CWI signaling pathway [35]. Normally, CWI is activated by heat shock stress or by agents that disrupt the cell wall [50]. Although Reinoso-Martín et al. [51] demonstrated that the functionality of CWI is related to tolerance to caspofungin action, when we used two mutant strains affected in different steps of CWI, we observed an increase in resistance to HAE. However, as reported by Parrish et al. [52], mutations in the myotubularin-related phosphatase Ymr1 and the synaptojanin-like phosphatases Sjl2 and Sjl3, essential in the regulation of phosphatidylinositol 3-phosphate (Ptdlns (3)P), cause a defect in the regulation of CWI, leading to problems in adaption after a heat shock, which is probably due to the higher levels of Slt2 phosphorylation. In addition, the activation of Pkc1-mediated CWI could contribute to the lethality effects caused by Ptdlns (3)P accumulation in phosphoinositide 3-phosphatase-deficient cells. This pathway can be activated through membrane lipids; for example, under heat stress, dihydrosphingosine and phytosphingosine increase in the membrane, which trigger the activation of the Pkh1 and Pkh2 kinases, which are essential for the activation of Pkc1 [35]. Therefore, it is likely that HAE may trigger the activation of the CWI signaling pathway through membrane lipids or by direct interaction with the Bck1, Mkk1, or Mkk2 proteins. This could explain the behavior observed with the mutants, since mutations in the *BCK1* or *MKK1* and *MKK2* genes do not allow for the normal functioning of the pathway, culminating in higher cell viability.

Strikingly, when *S. cerevisiae* cells were challenged with HAE in the presence of the osmotic stressor NaCl, viability was significantly decreased (Figure 3). It is known that cell wall deficiency renders yeast cells osmotically sensitive [53]; therefore HAE is also likely to directly affect the cell wall. Strikingly, *D. corticola* also exhibited exacerbated osmotic sensitivity in the presence of HAE (Figure 4). To our knowledge, studies on osmotic stress with *D. corticola* have not been conducted so far, but the resemblance with the *S. cerevisiae* model is compelling evidence of similar behaviors between both organisms. The evidence presented here clearly points to the cell wall as a target with a possible involvement in effects on the plasma membrane; the complexity of the mechanism of action of HAE is probably a result of the effect of the activity of several phytochemicals of the extract acting simultaneously and possibly in synergism.

## 4. Materials and Methods

### 4.1. Plant Collection and Preparation of the Extract

Leaves of *H. helix* were collected in October 2020 from an adult plant in a private garden in Braga, Portugal. All the leaves were air dried at room temperature, in the dark, for approximately four weeks. Subsequently, the leaves were frozen in liquid nitrogen and grounded into a fine powder with a pestle and mortar. The powdered plant material was extracted with deionized water at 100 °C, in a proportion of 1:10 (*w*/*v*), for 30 min and filtered through Whatman filter paper No. 1, yielding the HAE. The extract was lyophilized and stored in the dark at room temperature (±25 °C) until use.

### 4.2. Microorganisms and Culture Conditions

The *Saccharomyces cerevisiae* strains used in this work were BY4741 (*MAT*a *his3Δ1 leu2Δ0 met15Δ0 ura3Δ0*) and the derived mutants *erg2* (BY4741 *YMR202w::kanMX4;* provided by Marie Kodedová and Hana Sychrová from the Czech Academy of Sciences, Prague, Czech Republic), *bck1* (BY4741 *YJL095w::kanMX4*), and *mkk1*/*mkk2* (BY4741 *YPL140c::kanMX4 mkk1::LEU2*). These strains were cultivated in YPDA medium (2% *w*/*v* peptone, Bacto^TM^, Thermo Fisher Scientific, Porto Salvo, Portugal; 2% *w*/*v* dextrose, Difco^TM^**,** Thermo Fisher Scientific, Porto Salvo, Portugal; 2% *w*/*v* agar, Labchem, Laborspirit, Loures, Portugal; and 1% *w*/*v* yeast extract, FisherBioreagents, Fisher Scientific, Porto Salvo, Portugal) at 30 °C for 2 days and then stored at 4 °C. *Diplodia corticola* was kindly provided by Ana Cristina Esteves (Centre for Environmental and Marine Studies, CESAM, University of Aveiro, Portugal) and was cultivated on Potato Dextrose Agar (PDA; Biolife, Milan, Italy) in the dark at 25 °C. For yeast liquid cultures, 1-2 colonies were used as inoculum to YPD (YPDA without agar), and the cultures were incubated at 30 °C, 200 revolutions per minute (rpm). Growth was monitored by optical density at 600 nm (OD_600_).

### 4.3. Chemical Analysis

A solution of ethanol–deionized water (dH_2_O; 80:20 *v*/*v*) was used to dissolve the extract; the solution was filtered with a 0.2-μm nylon syringe filter and Dionex Ultimate 3000 UPLC was used for analysis (Thermo Scientific, San Jose, CA, USA). According to Barros et al. [53], the sample was injected using an automatic injector set to 5 °C, complete with a degasser and a column compartment thermostated to 35 °C. Compounds were found at 280, 330, and 370 nm using a diode array detector (DAD). A Waters Spherisorb S3 ODS-2 C18 column (4.6 × 150 mm; 3 μm; Milford, CT, USA) was used for compound separation in the reverse phase. The mobile phase consisted of (A) formic acid (0.1%) and (B) acetonitrile at a flow rate of 0.5 mL/min. The gradient elution regime included 10 to 15% B for 5 min, 15 to 20% B for 5 min, 20 to 25% B for 10 min, 25 to 35% B for 10 min, 35 to 50% B for 10 min, and rebalancing the column for 10 min. The system was adapted to an Ion Trap Linear LTQ XL mass spectrometer (Thermo Finnigan, San Jose, CA, USA) with an electrospray ionization (ESI) source with nitrogen as the carrier gas at 50 psi and a spray voltage of 5 kV. The initial temperature was 325 °C with a capillary voltage of −20 V and tube lens voltage of −66 V. Spectra of the charge mass ratio (*m*/*z*) were obtained in the negative mode, with a constant collision energy of 35 arbitrary units, between 100 and 1500. Data collection, processing, and analysis were performed using Xcalibur software, version 2.2, from Thermo Finnigan, San Jose, CA, USA. Retention time (Rt), wavelength of maximum absorption (λ_max_), mass spectra, pseudomolecular ion ([M-H]^−^), UV–Vis spectra, retention time, and patterns of ion breakdown (MS^2^) were compared with published research and commercially accessible standards in order to identify the chemicals [54].

### 4.4. Evaluation of Antifungal Activity In Vitro

The antifungal activity of HAE against *D. corticola* was assessed in vitro by measurements of the diameter of the mycelium with time. The extract was dissolved in dH_2_O, filter-sterilized, incorporated into a PDA medium after autoclaving, and cooled down to ~50 °C. A stock solution of 2.5 mg/mL of HAE was used for incorporation into the PDA for a 50 µg/mL final concentration, and a stock solution of 50 mg/mL HAE was used for incorporation into the PDA for 100, 500, 1000, and 1500 µg/mL. The mixtures were gently mixed by agitation, poured into Petri dishes, and left at room temperature to solidify. After solidification, a small portion (5 mm diameter) of fungal mycelium was excised from the margins of the 12-day old cultures and placed in the middle of the Petri dishes and the plates were maintained at 25 °C, protected from light. The negative control was prepared with sterilized deionized water instead of the extract in the same volume of the extract used for the treatment with the highest concentration. The assay was concluded when in the control plate, the mycelium occupied nearly all the medium surface (around 6 days of incubation). After 3 and 6 days of incubation, the mycelial colony diameter was measured, and the percentage of growth inhibition was determined using the following formula:$$Percentage\;of\;growth\;inhibition\;(\%)=\;\frac{dc-dt}{dc}\;\times \;100$$
where $d_{c}$ corresponds to the average mycelial diameter in the control and $d_{t}$ is the mycelial diameter in each replicate of each treatment. The percentage of inhibition was calculated for each replicate.

### 4.5. Assessment of Yeast Viability

The initial inoculum was prepared by taking 1-2 colonies of the selected strain (*Saccharomyces cerevisiae* BY4741) from the solid YPDA medium and inoculating them into fresh liquid YPD. Following overnight incubation, the suspension was diluted with fresh YPD medium to an OD_600_ = 0.1 and incubated for a further 4 h under the same conditions. The suspension was divided into aliquots once OD_600_ reached 0.4. Each aliquot corresponded to a different treatment with a different extract concentration: 10, 50, 75, 100, or 250 µg/mL. The negative control was prepared by replacing the extract volume by the solvent (sterilized dH_2_O) at the highest volume of extract used. All suspensions were incubated under the same conditions (30 °C, 200 rpm) and after 0, 30, 60, and 90 min, an aliquot of 100 µL was collected and serially diluted in sterilized dH_2_O to 10^−4^. Three 40 µL-drops of each dilution were placed on YPDA and incubated at 30 °C for 48 h. For each concentration of extract, the percentage of viability (percentage of colony-forming units; %CFUs) was calculated using the following equation:$$\%\;viability=\;\frac{{Mc}_{T}\times 100}{{Mc}_{T0}}$$
where, *M_cT_* corresponds to the average number of colonies counted at each time (30, 60, or 90 min) and *M_cT_*_0_ corresponds to the average number of colonies at time 0.

### 4.6. Evaluation of Osmotic Stress Susceptibility

The osmotic stress resistance of *S. cerevisiae* BY4741 was assayed with NaCl concentrations below and above 0.5 M [55]. In addition to the experimental condition without salt, three concentrations of the salt were tested—0.25, 0.5, and 0.75 M—and the influence of the extract, NaCl, and the combination of both on CFU was assessed. To prepare the cultures, the procedure described in the previous section (Section 4.5) was followed, in this case using Petri dishes with the salt incorporated in the YPDA medium. Briefly, on the day before the assay, the inoculum and the culture at OD_600_ = 0.4 were prepared as described above (Section 4.5). Two aliquots were used for the different HAE concentrations tested: 10 or 50 µg/mL, and a third aliquot was used as a negative control (the volume of extract was replaced by the solvent, sterilized dH_2_O, at the highest volume of extract used). After 0, 30, and 60 min at 30 °C and 200 rpm, an aliquot was collected, serially diluted in sterilized dH_2_0 to 10^−4^, and three 40 µL drops of each dilution were placed on YPDA plates with different concentrations of NaCl (0, 0.25, 0.5, or 0.75 M). Plates were incubated at 30 °C for 48 h, colonies were counted, and the percentage of cell viability was calculated using the same equation as in Section 4.5, but this time the *M_cT_* corresponded to the average number of colonies counted at each time (30, 60, and 90 min) on each concentration of extract and salt and the *M_cT_*_0_ corresponded with the average number of colonies at time 0 for each concentration of extract and salt. For the osmotic sensitivity assays with *D. corticola*, a similar procedure was adopted regarding the evaluation of antifungal activity in vitro (Section 4.4), with the exception of the preparation of PDA media with 0.4 M NaCl with and without HAE.

### 4.7. Statistical Analysis

The results were statistically analyzed using different statistical tests using GraphPad Prism version 8.0 for Windows, GraphPad Software (San Diego, CA, USA). A two-way ANOVA was used for in vitro assays, a one-way ANOVA and Dunnett’s post hoc test were used in the case of viability tests with *S. cerevisiae*, and a two-way ANOVA and Tukey’s post hoc test were used on the osmotic stress studies. The data were presented as mean ± standard deviation (SD) and obtained from at least three independent experiments in the case of viability tests with *S. cerevisiae* and in vitro assays.

## 5. Conclusions

In this work, we reported the activity of an aqueous extract made from *H. helix* leaves as an antifungal agent against *D. corticola*. To our knowledge, this is the first report of the activity of the *H. helix* extract against this devastating fungus to important crops. In addition, as far as we know, the mechanism of action was investigated for the first time. The fact that the cell wall is targeted by HAE ensures the specificity of toxic activity against fungi without affecting plants or animals. The remarkable activity of HAE provides the opportunity for the development of biodegradable and environmentally safe antifungal agents to combat the fungus *D. corticola* and consequently to minimize the negative economic impacts associated with the decline of cork oaks. It should also be noted that the extract, being aqueous, does not raise additional concerns for hypothetical applications in real natural conditions, since it does not contain solvents that are potentially dangerous to the ecosystem.

## Figures and Tables

**Figure 1 antibiotics-13-01116-f001:**
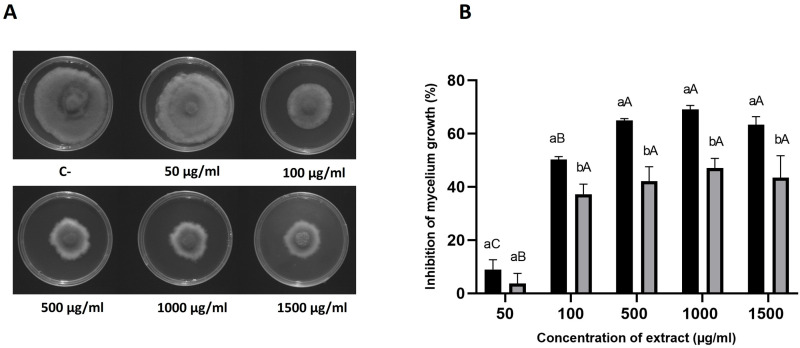
Percentage of growth inhibition of *D. corticola* by *H. helix* aqueous extract (HAE). Radial growth of *D. corticola* mycelium was measured in Petri dishes with PDA medium containing 50, 100, 500, 1000, or 1500 µg/mL of HAE. Representative images of HAE antifungal activity against *D. corticola* after 6 days of incubation (**A**). In the negative control (C-), the highest volume of extract used on the assays was replaced by sterilized deionized water (extract solvent). The diameter was measured after 3 (black bars; (**B**)) and 6 (grey bars; (**B**)) days of incubation at 25 °C in the dark. Each bar represents the mean ± SD of three independent experiments. A two-way ANOVA was used for the analysis and different letters represent statistical significance. Capital letters were used for comparisons of extract concentrations within the same time of incubation and lowercase letters were used for comparisons of different times of incubation for each concentration tested.

**Figure 2 antibiotics-13-01116-f002:**
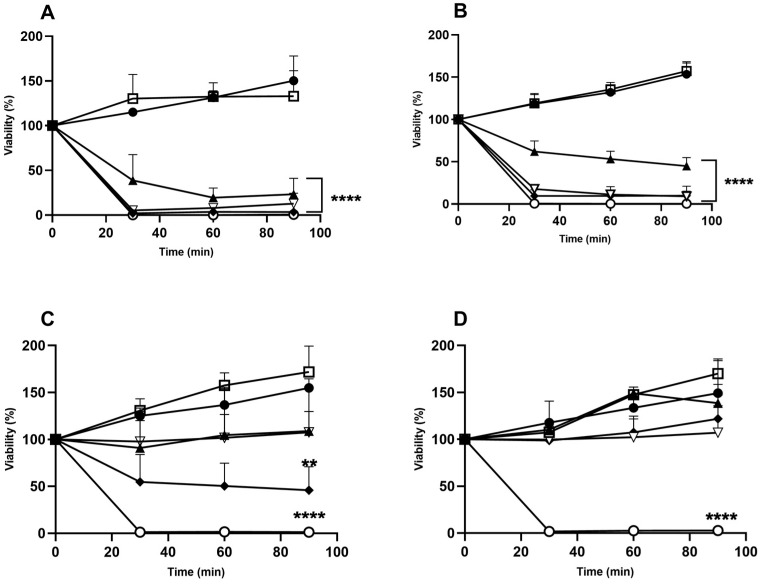
Viability of *S. cerevisiae* BY4741 (**A**) and the mutant strains *erg2* (**B**), *bck1* (**C**), and *mkk1/mkk2 (***D**) in the presence of aqueous extract of *H. helix* (HAE). Cells from exponentially growing cultures were exposed to 10 µg/mL (white squares), 50 µg/mL (black triangles), 75 µg/mL (white inverted triangles), 100 µg/mL (black diamonds), or 250 µg/mL (white circles) of HAE and incubated at 30 °C, 200 rpm. Viability was assessed using CFU after 0, 30, 60, and 90 min of incubation upon spreading 10^−4^ dilutions on YPDA plates and incubation at 30 °C, 200 rpm. The negative control (black circles) was prepared by replacing the extract by the solvent at the highest volume of extract used in the assays. The data represent the mean ± SD of three independent experiments. A one-way ANOVA and Dunnett’s post hoc test were used for the analysis, and concentrations were compared at each time-point. For clarity of graphical representation, significance is shown only for the 90 min timepoint, where ** means 0.001 < *p* ≤ 0.01 and **** means *p* < 0.0001; the absence of significance is not marked with any symbol. Other relevant significances are presented in the text.

**Figure 3 antibiotics-13-01116-f003:**
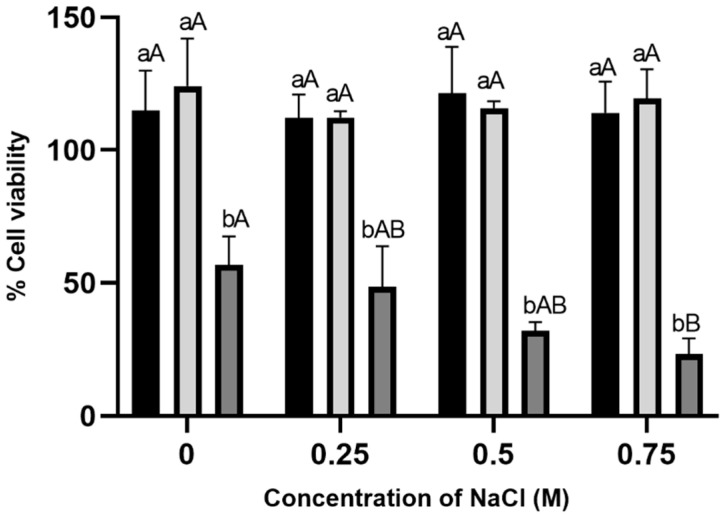
Viability of *S. cerevisiae* BY4741 in the presence of aqueous extract of *H. helix* (HAE) and osmotic stress. Cells from exponentially growing cultures were exposed to 10 µg/mL (grey bars) or 50 µg/mL (dark grey bars) of HAE at 30 °C, 200 rpm, for 30 min. The viability was assessed using CFU by plating 10^−4^ dilutions on YPDA plates containing different concentrations of NaCl (0, 0.25, 0.5, or 0.75 M) and further incubation at 30 °C for 48 h. The negative control (black bars) was prepared by replacing the extract with the solvent at the highest volume of extract used in the assays. The data represent the mean ± SD of three independent experiments. A two-way ANOVA and Tukey’s post hoc test were used for the analysis. A letter code was used. Lowercase letters are used for comparisons of extract concentration effects within each salt concentration (*p* < 0.0001) and capital letters are used for comparisons between salt concentrations for each extract concentration (*p* < 0.01). Mean values followed by the same letters are not statistically different.

**Figure 4 antibiotics-13-01116-f004:**
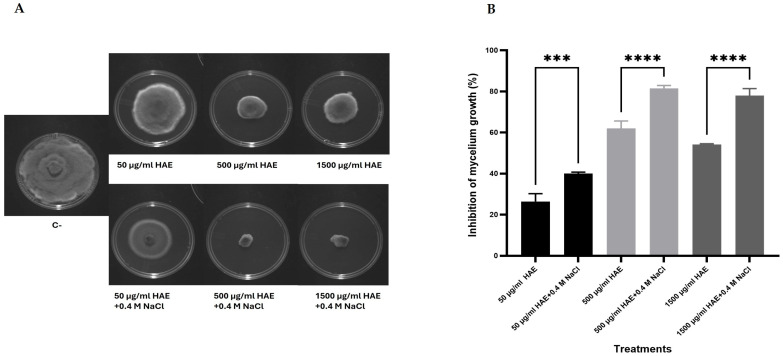
Percentage of growth inhibition of *D. corticola* by *H. helix* aqueous extract (HAE) in the absence and presence of osmotic stress. Radial growth of *D. corticola* mycelium was measured in Petri dishes with PDA medium containing 50, 500, or 1500 µg/mL HAE in the absence or presence of 0.4 M of NaCl. Representative images of HAE antifungal activity against *D. corticola* after 6 days of incubation (**A**). In the negative control (C-), the highest volume of extract used on the assays was replaced by sterilized deionized water (extract solvent). The diameter was measured after 6 days of incubation at 25 °C in the dark and the percentage of growth inhibition was calculated, taking C- as a reference (**B**). Each bar represents the mean ± SD of three independent experiments. A one-way ANOVA and Dunnett’s post hoc test were used for the analysis, where *** means 0.0001 < *p* ≤ 0.001 and **** means *p* < 0.0001.

**Table 1 antibiotics-13-01116-t001:** Chemical composition of aqueous extract of *H. helix* determined by electrospray ionization–mass spectrometry in the negative ionization mode [M-H].

Peaks	Rt	λmax	[M-H] *m*/*z*	MS^2^	Tentative Identification	Quantification (mg/g)
**1**	4.34	331	191	161 (100)	Quinic acid	3.42 ± 0.04
**2**	5.79	322	353	191 (100)	5-*O*-Caffeoylquinic acid	1.87 ± 0.01
**3**	6.03	329	431	341 (24), 311 (100)	Apigenin-8-*C*-hexoside	0.26 ± 0.01
**4**	6.41	280sh326	539	353 (56), 191 (21)	Yunnaneic acid D	0.73 ± 0.02
**5**	7.54	339	577	487 (41), 473 (3), 457 (100), 353 (38)	Apigenin-6-*C*-hexoside-8-*C*-rhamnoside	0.31 ± 0.01
**6**	6.99	326	497	353 (46), 337 (28), 335 (62), 211 (28), 179 (100)	Di-caffeoylshikimic acid	1.74 ± 0.03
**7**	13.19	335	463	287 (100)	Eriodictyol-*O*-glucuronide	1.87 ± 0.05
**8**	16.61	351	609	301 (100)	Quercetin-3-*O*-rutinoside	1.72 ± 0.02
**9**	19.74	331	593	285 (100)	Kaempferol-3-*O*-rutinoside	1.15 ± 0.03
**10**	20.71	344	623	315 (100)	Isorhamnetin-*O*-rutinoside	0.82 ± 0.04
**11**	30.11	283	1219	423 (100), 367 (32)	Hederacoside C	2.21 ± 0.01
					Total Phenolic Compounds	16.1 ± 0.3
					Total Flavonoids	6.1 ± 0.2
					Total Flavone	0.57 ± 0.02
					Total Flavanone	1.87 ± 0.05
					Total Flavonol	3.7 ± 0.1
					Total Phenolic Acid	7.8 ± 0.1
					Saponin	2.21 ± 0.01

Retention time (Rt), maximum visible absorption wavelengths (λmax), pseudomolecular and mass spectral fragment ions (in brackets, relative abundances), and possible identification and quantification of components in the examined aqueous extract of *H. helix* were calculated. sh: shoulder.

## Data Availability

The raw data supporting the conclusions of this article will be made available by the authors on request.

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
