# Peer review of "Cell Wall-Mediated Antifungal Activity of the Aqueous Extract of *Hedera helix* L. Leaves Against *Diplodia corticola"

_antibiotics, 2024, doi:10.3390/antibiotics13121116_

Round 1
Reviewer 1 Report
Comments and Suggestions for Authors
The aim of this article is to investigate the antifungal activity of aqueous extracts of H. helix (HAE) against D. corticola as a possible tool in the control of this fungal infection of cork oak. In vitro experiments demonstrated the activity of ivy extract against D. corticola growth with 70% and 45% inhibition after 3 and 6 days of incubation, respectively. To investigate the mechanism of action of HAE, the authors conducted experiments using S. cerevisiae to determine whether the plasma membrane or the cell wall were potential targets for HAE compounds. All S. cerevisiae kinase mutants were more resistant to HAE than the wild type. Furthermore, when S. cerevisiae cells were exposed to HAE in the presence of the osmotic stressor NaCl, their viability was significantly reduced. In conclusion, the authors suggest that at least one of the targets of HAE may be the cell wall.
There are some issues and drawbacks to be pointed out:
Major issues:
Data on the comparison of ethanol (e.g. paper from 2017 - doi: 10.1556/018.68.2017.2.7) with your water extracts in terms of constituents found are missing, as well as any rationale for using the water extract against D. corticola. These points should be addressed and clarified in the revised manuscript.
The use of S. cerevisiae as a model organism to elucidate the mechanism of action of HAE on D. corticola appears to be questionable. The results are not readily transferable between unrelated classes of fungi, namely Dothideomycetes and Saccharomycetes.
In order to validate the hypothesis that the cell wall is a target of HAE (based on data with S. cerevisiae), it is necessary to determine the viability of D. corticola in the presence of the osmotic stressor (NaCl). Otherwise, the data obtained from S. cerevisiae as a model organism are unreliable and any conclusions drawn from it may be misleading.
Minor issues:
Line 137: “peak 8” – should be in bold
Line 160: Table 1 requires a new caption to be added. The existing text is a legend and should be placed below the table. The peak # should be included, and the abbreviation "280sh326" requires elucidation (it may be "shoulder").
Line 178: Please clarify what is meant by 100% inhibition in Figure 1B. Please clarify which “extract solvent” was utilized.
Line 187: “tion” – It should be corrected to read "inhibition."
Line 190: It would be beneficial to include a functional description of the mutants used, as this information is currently only available in the discussion section, which is too late for inclusion.
In conclusion, the paper should be completed with data on the viability of D. corticola in the presence of the osmotic stressor, which will allow for a more comprehensive analysis of the subject matter. Some minor corrections are also required. The article is not yet suitable for publication in its current form.
Author Response
Comments and Suggestions for Authors
The aim of this article is to investigate the antifungal activity of aqueous extracts of H. helix (HAE) against D. corticola as a possible tool in the control of this fungal infection of cork oak. In vitro experiments demonstrated the activity of ivy extract against D. corticola growth with 70% and 45% inhibition after 3 and 6 days of incubation, respectively. To investigate the mechanism of action of HAE, the authors conducted experiments using S. cerevisiae to determine whether the plasma membrane or the cell wall were potential targets for HAE compounds. All S. cerevisiae kinase mutants were more resistant to HAE than the wild type. Furthermore, when S. cerevisiae cells were exposed to HAE in the presence of the osmotic stressor NaCl, their viability was significantly reduced. In conclusion, the authors suggest that at least one of the targets of HAE may be the cell wall.
There are some issues and drawbacks to be pointed out:
Major issues:
Data on the comparison of ethanol (e.g. paper from 2017 - doi: 10.1556/018.68.2017.2.7) with your water extracts in terms of constituents found are missing, as well as any rationale for using the water extract against D. corticola. These points should be addressed and clarified in the revised manuscript.
Authors response:
Thank you for your valuable feedback. In response to your comment, we have now added a comparison between ethanol-based extracts and our aqueous extract, highlighting differences in phenolic compounds, particularly in saponins and flavonoids (see lines 176-185 in the revised version of the manuscript). We have also clarified our rationale for using the aqueous extract against Diplodia corticola. This choice is based on the extract’s physiological compatibility and traditional use, as well as its demonstrated antimicrobial activity.
The use of S. cerevisiae as a model organism to elucidate the mechanism of action of HAE on D. corticola appears to be questionable. The results are not readily transferable between unrelated classes of fungi, namely Dothideomycetes and Saccharomycetes.
In order to validate the hypothesis that the cell wall is a target of HAE (based on data with S. cerevisiae), it is necessary to determine the viability of D. corticola in the presence of the osmotic stressor (NaCl). Otherwise, the data obtained from S. cerevisiae as a model organism are unreliable and any conclusions drawn from it may be misleading.
Authors response:
We appreciate the Reviewer’s suggestion. We have performed additional experiments to investigate the osmotic sensitivity of D. corticola im the absence and presence of the extract. The results are presented in Fig. 4, in lines 320-329, discussed in lines 443-446 and the procedure was added in the Materials and Methods section (lines 557-560).
Minor issues:
Line 137: “peak 8” – should be in bold
Authors response:
The manuscript was changed accordingly (line 165 in the revised manuscript).
Line 160: Table 1 requires a new caption to be added. The existing text is a legend and should be placed below the table. The peak # should be included, and the abbreviation "280sh326" requires elucidation (it may be "shoulder").
Authors response:
Table 1 was changed accordingly (see title and legend of Table 1 in the revised manuscript).
Line 178: Please clarify what is meant by 100% inhibition in Figure 1B. Please clarify which “extract solvent” was utilized.
Authors response:
100% inhibition means the ability of the aqueous extract of Hedera helix to completely inhibit the growth of the fungus compared to the control (absence of extract). The percentage of inhibition was calculated according to the equation presented in the section 4.4 of Materials and Methods. The solvent of the extract was sterilised deionised water. In the legend of Figure 1 the solvent of the extract was mentioned explicitly.
Line 187: “tion” – It should be corrected to read "inhibition."
Authors response:
This is a typo that was overlooked in the revisions of the manuscript before submission. We thank the Reviewer’s suggestion of “inhibition”, however the original word is “action”. We have decided to keep the original word (see line 213 in the revised manuscript).
Line 190: It would be beneficial to include a functional description of the mutants used, as this information is currently only available in the discussion section, which is too late for inclusion.
Authors response:
In accordance with the Reviewer’s suggestion, the mansucript was changed by including the description of the mutant strains and their relevant phenotypes (see lines 218-228 of the revised manuscript).
Reviewer 2 Report
Comments and Suggestions for Authors
This study explores the antifungal activity of Hedera helix leaves aqueous extract (HAE) against D. corticola. Chemical analysis revealed antimicrobial compounds such as hederacoside C, quinic acid, and flavonoid glycosides. HAE reduced radial mycelial growth by 70% in vitro. Mutant Saccharomyces cerevisiae strains with disrupted cell wall integrity pathways were more resistant to HAE, indicating that the extract targets these pathways.
1. Authors have performed good work however the discussion part need to be revised as it is very lengthy and include repetition of same things at many place.
Author Response
Comments and Suggestions for Authors
This study explores the antifungal activity of Hedera helix leaves aqueous extract (HAE) against D. corticola. Chemical analysis revealed antimicrobial compounds such as hederacoside C, quinic acid, and flavonoid glycosides. HAE reduced radial mycelial growth by 70% in vitro. Mutant Saccharomyces cerevisiae strains with disrupted cell wall integrity pathways were more resistant to HAE, indicating that the extract targets these pathways.
- Authors have performed good work however the discussion part need to be revised as it is very lengthy and include repetition of same things at many place.
Authors response:
The discussion section was reformulated with exclusion of redundant parts (see MSWord changes tracking in the Discussion of the revised manuscript).
Reviewer 3 Report
Comments and Suggestions for Authors
The authors of the manuscript "Cell wall-mediated antifungal activity of the aqueous extract of Hedera helix leaves against Diplodia corticola" present an intriguing manuscript concerning the extraction, characterization, and antifungal activity of aqueous extracts of H. helix against in vitro cultures of D. corticola. The manuscript also presents a study of a potential mechanism of action for HAE, utilizing mutants of Saccharomyces cerevisiae that have been extensively characterized for their metabolic properties, particularly in structures such as the plasma membrane and cell wall. To substantiate the findings, tests based on osmotic stress resistance were conducted on mutant and non-mutant S. cerevisiae cultures.
The manuscript is clearly written and presents interesting results. However, there are a few minor issues that require attention.
Please address the following points:
Abstract - Line 28: Please insert a symbol (perhaps “/”?) between the two mutated genes. Please have the authors check this.
Keywords - Line 34: keywords should be revised to remove "ivy." H. helix is already referenced in the line, and its common name is provided in the introduction.
Introduction - lines 100-102: Please provide a more focused overview of the purpose of the work in a separate paragraph. Now, the text is discursive and seems to serve merely as an appendix to the introduction.
Results - section 2.1 - lines 147-158: why did the authors include this paragraph downstream of the identification (lines 105-146)? Shouldn't it be the other way around? First this paragraph and then the peak analysis with identification. The first sentence (lines 147-148) seems to endorse my impression. I kindly request that the authors review and implement the necessary corrections.
Results - section 2.1 - line 149: The authors should specify in the quantification whether they are talking about fresh weight (FW) or dry weight (DW). The authors are kindly requested to review all other concentrations.
Table 1 - line 162: Please use the abbreviated form H. helix. The specie is already mentioned in the text.
Results - Legend Figure 2: a) the species names may be written in abbreviated form, as they have already been cited; b) for accuracy and clarity, please indicate the level of significance in the legend; c) please provide an explanation as to why a two-way ANOVA test was utilized instead of, for instance, a one-way ANOVA and Tukey's HSD test between 3 and 6 days.
Results - section 2.3 - line 187: please review the title of section 2.3.
Results – Figure 3: can figure 3 be simplified? Too many letters can be confusing...
Materials and Methods - section 4.4 - lines 415-418: How was HAE incorporated? Was the alcohol extract used? How was it evenly distributed in the sterilized medium? Perhaps it would be good to write down the HAE concentration to get an idea of the volumes involved.
Author Response
Comments and Suggestions for Authors
The authors of the manuscript "Cell wall-mediated antifungal activity of the aqueous extract of Hedera helix leaves against Diplodia corticola" present an intriguing manuscript concerning the extraction, characterization, and antifungal activity of aqueous extracts of H. helix against in vitro cultures of D. corticola. The manuscript also presents a study of a potential mechanism of action for HAE, utilizing mutants of Saccharomyces cerevisiae that have been extensively characterized for their metabolic properties, particularly in structures such as the plasma membrane and cell wall. To substantiate the findings, tests based on osmotic stress resistance were conducted on mutant and non-mutant S. cerevisiae cultures.
The manuscript is clearly written and presents interesting results. However, there are a few minor issues that require attention.
Please address the following points:
Abstract - Line 28: Please insert a symbol (perhaps “/”?) between the two mutated genes. Please have the authors check this.
Authors response:
A slash has been inserted between the recessive alleles mkk1 and mkk2 in the double mutant strain (see line 28 in the revised manuscript).
Keywords - Line 34: keywords should be revised to remove "ivy." H. helix is already referenced in the line, and its common name is provided in the introduction.
Authors response:
We have removed the word “ivy” from the keywords list (see line 34 in the revised manuscript highlighted with MSWord changes tracking).
Introduction - lines 100-102: Please provide a more focused overview of the purpose of the work in a separate paragraph. Now, the text is discursive and seems to serve merely as an appendix to the introduction.
Authors response:
We acknowledge the discursive mode of the description of the purpose of the work in the original version of the manuscript and thank the Reviewer for pointing it out. The objective of the work was reformulated accordingly and was placed in a single paragraph at the end of the Introduction (see lines 101-111 of the revised version of the manuscript).
Results - section 2.1 - lines 147-158: why did the authors include this paragraph downstream of the identification (lines 105-146)? Shouldn't it be the other way around? First this paragraph and then the peak analysis with identification. The first sentence (lines 147-148) seems to endorse my impression. I kindly request that the authors review and implement the necessary corrections.
Authors response:
Thank you for your valuable comments. We have addressed your concern regarding the order of the paragraph in Section 2.1. The paragraph discussing the quantification of compounds is now before the paragraph of the peak identification, as suggested (see lines 115-133 in the revised manuscript). Additionally, we have ensured a clear transition between these sections to improve the flow of information. We hope these changes meet your expectations.
Results - section 2.1 - line 149: The authors should specify in the quantification whether they are talking about fresh weight (FW) or dry weight (DW). The authors are kindly requested to review all other concentrations.
Authors response:
Reference to fresh or dry weight was not included because the quantification of the compounds was made in relation to lyophilised extract. So, the concentration of the compounds should be read mg of compound per g of lyophilised extract.
Table 1 - line 162: Please use the abbreviated form H. helix. The specie is already mentioned in the text.
Authors response:
We adopted the rule that the scientific names in the beginning of sentences and in table or figure legends should not be abbreviated. We understand the concern of Reviewer 3, we have changed the name to the abbreviated form as suggested (see title and footnote of Table 1).
Results - Legend Figure 2: a) the species names may be written in abbreviated form, as they have already been cited; b) for accuracy and clarity, please indicate the level of significance in the legend; c) please provide an explanation as to why a two-way ANOVA test was utilized instead of, for instance, a one-way ANOVA and Tukey's HSD test between 3 and 6 days.
Authors response:
- a) As we mentioned above, non-abbreviated forms of scientific names were adopted whenever they were located in the beginning of sentences or in legends of figures or titles of tables. We have changed all species names in accordance to th Reviewer’s suggestion (see legends of all figures and the title of Table 1).
- b) The legend was changed according to the Reviewer’s suggestion. Significance was added also to the graphs but only to the 90 min points for the sake clarity of the graphical representation, while relevant comparisons are described in the text as in the first version of the manuscript.
- c) Reviewer 3 probably refers to Fig. 1B since this is the only graph where measurements after 3 and 6 days are depicted. Here we used a two-way ANOVA (followed by multiple comparisons Tukey’s test), instead of the one-way, because we aimed to analyse the effect of both factors: extract concentration and the inhibitory response along time. As shown in Fig. 1B, there’s a dose-dependent increase in inhibition of radial growth up to 500 µg/mL, plateauing from there to the highest concentration tested (1500 µg/mL), at day 3. Also, a significant reduction in percentage inhibition was observed from day 3 to day 6 in all tested concentrations except the lowest one (almost without impact on mycelial growth), with no significant differences in percentage inhibition.
Results - section 2.3 - line 187: please review the title of section 2.3.
Authors response:
This was a typo that was overlooked in the final revisions before submission of the manuscript. The title was changed to the correct form (see line 213 of the revised manuscript).
Results – Figure 3: can figure 3 be simplified? Too many letters can be confusing...
Authors response:
We understand the concern of Reviewer 3. We wanted to show the impact of the extract under each condition of osmotic challenge (lowercase letters) and the impact of the osmotic stress under each condition of exposure to the extract (uppercase letters). We concluded that the letter code for the significance was the simplest one to show both statistical analyses in the same graph. In some cases, three letters are represented because differences with other conditions are significant and non-significant. We believe that, although the representation seems complex, this format allows to clearly show all statistical analyses in a correct and formal way.
Materials and Methods - section 4.4 - lines 415-418: How was HAE incorporated? Was the alcohol extract used? How was it evenly distributed in the sterilized medium? Perhaps it would be good to write down the HAE concentration to get an idea of the volumes involved.
Authors response:
The extract HAE used in this work was a water extract. HAE was filter sterilised and incorporated into PDA medium after autoclaving and cooled down to ~50 °C. The mixture was gently mixed by agitation, poured into Petri dishes and left at room temperature to solidify. For the final concentration of 50 µg/ml, a 2.5 mg/ml HAE stock solution was used (400 µl HAE per Petri dish with 20 ml PDA) and for the higher concentrations a 50 mg/ml HAE stock solution was used. So, for 100 µg/ml HAE final concentration, 40 µl of stock were added per Petri dish; for 500 µg/ml, 200 µl HAE were added; for 1000 µg/ml, 400 µl HAE were added; and for 1500 µg/ml we added 600 µl. In Materials and Methods section we added a brief description of the incorporation of HAE into PDA (see lines 504-509 of the revised manuscript).
Round 2
Reviewer 1 Report
Comments and Suggestions for Authors
The manuscript has been sufficiently improved to warrant publication in Antibiotics.
Author Response
The manuscript has been sufficiently improved to warrant publication in Antibiotics.
Response
We thank the Reviewer's comment and the Reviewer's contributions for the improvement of the manuscript.